# Using shape to turn off blinking for two-colour multiexciton emission in CdSe/CdS tetrapods

Nimai Mishra[1,*], Noah J. Orfield[1,*], Feng Wang[1], Zhongjian Hu[1], Sachidananda Krishnamurthy[1,2], Anton V. Malko[2], Joanna L. Casson[3], Han Htoon[1], Milan Sykora[3] & Jennifer A. Hollingsworth[1]

Semiconductor nanostructures capable of emitting from two excited states and thereby of producing two photoluminescence colours are of fundamental and potential technological significance. In this limited class of nanocrystals, CdSe/CdS core/arm tetrapods exhibit the unusual trait of two-colour (red and green) multiexcitonic emission, with green emission from the CdS arms emerging only at high excitation fluences. Here we show that by synthetic shape-tuning, both this multi-colour emission process, and blinking and photobleaching behaviours of single tetrapods can be controlled. Specifically, we find that the properties of dual emission and single-nanostructure photostability depend on different structural parameters—arm length and arm diameter, respectively—but that both properties can be realized in the same nanostructure. Furthermore, based on results of correlated photoluminescence and transient absorption measurements, we conclude that hole-trap filling in the arms and partial state-filling in the core are necessary preconditions for the observation of multiexciton multi-colour emission.

[1] Materials Physics and Applications Division, Center for Integrated Nanotechnologies, Los Alamos National Laboratory, PO Box 1663, MS-K771, Los Alamos, New Mexico 87545, USA. [2] Department of Physics, The University of Texas at Dallas, Richardson, Texas 75080, USA. [3] Chemistry Division, Los Alamos National Laboratory, Los Alamos, New Mexico 87545, USA. * These authors contributed equally to this work. Correspondence and requests for materials should be addressed to M.S. (email: sykoram@lanl.gov) or to J.A.H. (email: jenn@lanl.gov).

CdSe/CdS core/arm tetrapods have been investigated for their unusual characteristics of ultra-large absorption cross-section[1], long biexciton lifetimes[2] and strain-dependent emission[3]. These traits render such quantum dot (QD)-'seeded' tetrapods potentially useful as efficient energy-conversion materials for solar-cell applications[1] or as low-threshhold gain media for lasing applications[2], respectively. Less well explored, however, is their tendency for exhibiting dual emission.

Dual or two-colour-emitting nanostructures are of fundamental interest as model systems for understanding exciton (bound electron–hole pair) generation and relaxation processes in quantum-confined systems. They are also technologically significant as nanoscale thermometers capable of highly sensitive ratiometric temperature sensing[4,5] and as potential new photon sources for white-light generation[6]. Two-colour photoluminescence (PL) from the same semiconductor nanostructure can be obtained, for example, from QDs that are chemically doped with transition metal impurity ions[4,5,7,8], or from complex nanocrystals in which a potential energy barrier layer is synthetically added to divide different parts of the nanostructure, such as a core QD and an outer shell (symmetric or asymmetric) separated by an electron and/or hole tunnelling barrier[9–13].

CdSe/CdS tetrapods are interesting dual-emitting nanostructures, as they can exhibit two types of dual emission behaviour—barrier-mediated and multiexcitonic dual emission, where the latter is distinct from the two-colour PL observed from other nanostructures. Barrier-mediated dual emission in tetrapods involves two red-emission bands associated with the CdSe core[14,15]. Specifically, in single-tetrapod studies conducted at cryogenic temperatures, two red-colour emissions were observed and attributed to emission from direct and indirect exciton states, respectively. The former involves radiative recombination of a CdSe-localized electron–hole pair, or exciton, whereas the latter involves recombination of either a CdSe-localized hole and a CdS-localized electron (type II band alignment) or a CdSe-localized hole and a delocalized electron (quasi type II band alignment)[15]. The ability of the direct exciton state to coexist with the indirect exciton state was attributed to the presence of a strain-induced interfacial barrier inhibiting electron relaxation to the CdS shell (type II structure) or delocalization into the CdS shell (quasi type II structure). The two red emissions (direct, 2.045 and indirect, 2.020 eV) were observed at all excitation pump fluences and afforded typical random spectral diffusion and fluorescence blinking behaviours[15].

In contrast, the second type of dual emission observed in QD-seeded tetrapods appears only at high excitation pump fluences. Under such conditions, green emission from the CdS arms can emerge in addition to CdSe-core red emission. This constitutes a multi-excitonic emission process as intense optical excitation results in the formation of multiple excitons that contribute together to the observed PL and, although the exciton states that produce these red and green emissions are spatially separated between the core and an arm, respectively, they are electronically coupled. This is in contrast with excitonic states responsible for dual emission in core/shell/shell QDs, which are both spatially and electronically decoupled[9,10]. In tetrapods, photoexcitation creates excitons in the core and the arms in proportion to their relative volumes and absorption cross sections. High-energy carriers (electrons and holes) generated during excitation in the arms can relax or migrate as excitons into the core-localized states, whereby tetrapod arms effectively function as a light-collecting antenna for the core. A previous study[16] has concluded that at sufficiently high pump levels core electronic states can get filled, thus blocking further relaxation of carriers (or migration of excitons) from the CdS arms to the core. Under

these conditions, the arm states can potentially contribute to tetrapod PL. Concurrent suppression of non-radiative Auger recombination (Auger recombination is a many-body process whereby the energy of electron-hole recombination is transferred to a separate electron or hole that then relaxes from its excited state to the ground state non-radiatively by dissipation of heat) was also cited as a prerequisite for this type of two-colour PL. Tetrapod volume was identified as the key structural feature determining whether Auger processes were sufficiently suppressed, with long-arm (55 nm) tetrapods affording sufficiently large volumes in contrast with short-arm (28 nm) tetrapods[16].

To date, multi-excitonic two-colour emission has been studied for ensembles of tetrapods[6,16]. Here we investigate room-temperature PL properties at the single tetrapod level. We study the blinking, photobleaching and quantum optical properties of the multiexcitonic dual-emission process, and we do so as a function of both tetrapod arm length and arm thickness. In this way, we are able to differentiate effects of nanostructure volume from those of tetrapod arm length or thickness. We also elaborate on the role of Auger recombination in this system, distinguishing between two types of Auger processes—those active in the core and those in the arms. We show that Auger recombination in the CdSe core determines the extent to which PL blinking can be suppressed, whereas Auger recombination in the CdS arms probably determines the prevalence of two-colour red and green emission. We further show that large volume is an insufficient condition for achieving the necessary suppression of Auger recombination in the arms for strong dual emission and, instead, tetrapod shape plays the key role, with arm thickness defining Auger processes in the core and arm length dominating processes in the arms. Significantly, we take advantage of recent advances in tetrapod synthesis, as well as exploiting alternative methods for tuning tetrapod arm thickness, to prepare a comprehensive tetrapod-shape series that allows the direct assessment of 'geometry' effects on single-tetrapod PL properties from photostability (blinking, bleaching) to colour complexity. In this way, we determine the structural parameters for intentionally 'engineering' unprecedented performance metrics of blinking-suppressed two-colour multiexcitonic and even tricolour PL.

## Results

**Synthesis and characterization of a tetrapod shape series.** Using 4 nm CdSe zinc-blende QDs[17] as seed material, we synthesized a series of tetrapods characterized by differences in both arm thickness and arm length. We employed the seeded growth approach of Fiore et al.[18] with modified surfactant chemistry for enhanced monodispersity[2,19,20]. The resulting tetrapods were as follows: thin/short (TP1: 6.3 nm ± 0.5 nm/24.8 nm ± 2.3 nm), thick/short (TP2: 10.9 nm ± 0.8 nm/27.6 nm ± 2.9 nm), thin/long (TP3: 8.2 nm ± 1.3 nm/40.7 nm ± 3.8 nm) and thick/long (TP4: 10.6 nm ± 2.4 nm/41.3 nm ± 4.6 nm) (Fig. 1). (Note: TP diameters were measured at the arm base.) TP1, TP3 and TP4 were prepared exclusively using the recently developed syntheses[2,19,20], whereas TP2 tetrapods were fabricated from TP1 tetrapods using the successive ionic layer adsorption and reaction (SILAR) method to controllably add CdS shell monolayers[21]. Despite the complex shape of the tetrapod TP1 'core', SILAR growth proceeded roughly as expected, with the number of predicted shell monolayers (calculated based on the amount of precursor added; Supplementary Note 1 in Supplementary Information) approximately matching the number of shell monolayers obtained in the synthesis, as determined by measuring the arm-thickness increase from transmission electron microscopy images. Specifically, nominally five mono-layers of CdS were added (one monolayer = 0.3375 nm) by the

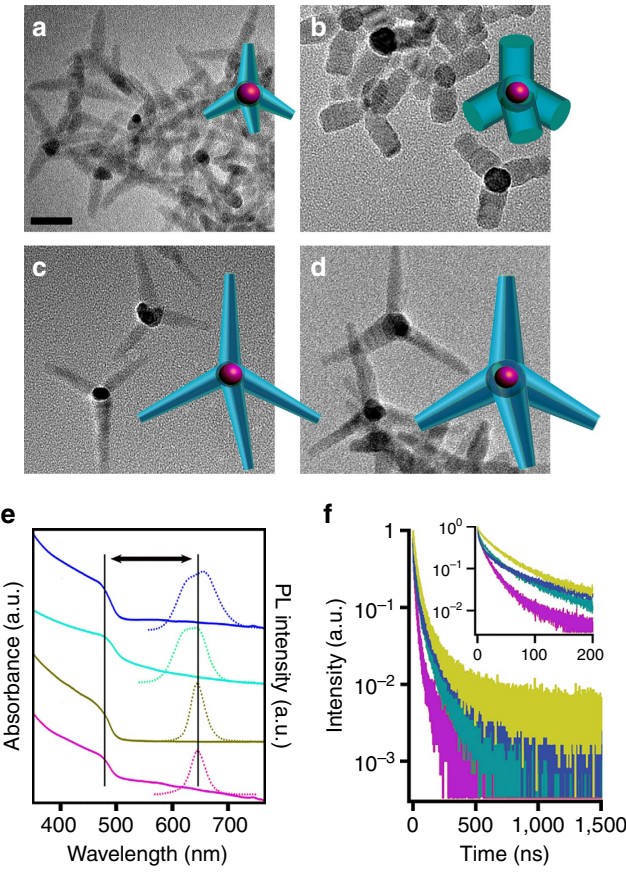

**Figure 1 | Electron microscopy images and ensemble optical properties for a nanoengineered tetrapod shape series.** (a–d) Transmission electron microscopy (TEM) images of four tetrapods (also shown schematically) differing in arm diameter (measured at the base of 3 arms for 30–40 tetrapods in each case) and arm length (also measured for 3 arms of 30–40 tetrapods for each of the tetrapod geometries), respectively— (**a**) TP1: 6.3 nm ± 0.5 nm and 24.8 nm ± 2.3 nm; (**b**) TP2: 10.9 nm ± 0.8 nm and 27.6 nm ± 2.9 nm; (**c**) TP3: 8.2 nm ± 1.3 nm and 40.7 nm ± 3.8 nm; (**d**) TP4: 10.6 nm ± 2.4 nm and 41.3 nm ± 4.6 nm (scale bar, 20 nm (in **a** applies also to **b**–**d**)) (**e**) Absorbance and PL spectra for each of the tetrapod geometries (TP1 (magenta), TP2 (ochre), TP3 (turquoise) and TP4 (blue)) showing the large effective Stokes shift between absorption positions and PL maxima (~160 nm; indicated by arrow). (**f**) Ensemble PL lifetime measurements for red emission (~650 nm) in the CdSe core (same colour scheme as **e**).

SILAR technique and the resulting arm diameter increased by ~4.1 nm compared with a predicted 3.4 nm. Interestingly, however, CdS deposition was not conformal. As-prepared, tetrapod arms adopt a truncated cone shape[2,19], but SILAR-thickened arms were cylindrical (Fig. 1a compared with Fig. 1b), implying that SILAR growth was more rapid at the thinner arm tips than at the thicker arm bases. In all cases, tetrapod volume is dominated by the relatively large CdS arms compared to the relatively small CdSe core. Specifically, the volume of the CdSe core is ~35 nm³, whereas total tetrapod volumes (calculated using dimensions measured from transmission electron microscopy images and the volume of a truncated cone as an estimate of arm shape for TP1, TP3 and TP4, and that of a cylinder for TP2; Supplementary Note 1) are much larger: ~2,660 nm³ (TP1), 10,330 nm³ (TP2), 6,840 nm³ (TP3) and 12,270 nm³ (TP4).

CdSe/CdS tetrapods exhibit a large effective Stokes shift[1] or energy separation between the onset of absorption and the PL peak position (Fig. 1e). The higher bandgap CdS arms contribute more to the total particle volume and thereby dominate absorption spectra, whereas PL derives from exciton recombination in the CdSe core, effectively separating the processes of absorption and emission. A significant red-shift in tetrapod PL (centred at ~645–655 nm) from that of the starting CdSe 'seed' QDs (605 nm) is also observed. A similar result is obtained for thick-shell 'giant' CdSe/CdS QDs as well as CdSe/CdS nanorods and attributed to the quasi type II core/shell electronic structure that is characteristic of the CdSe/CdS system[1,22–25]. In contrast to a type I arrangement of core and shell conduction and valence band levels, in a quasi type II structure, the electronic wavefunction is not completely confined to the core[23,24], affording partial electron–hole spatial separation and the observed PL red shifting. The strong influence of the quasi type II band alignments on PL energies is in contrast with single-composition tetrapods, for example, CdTe tetrapods, whose PL energies depend instead on quantum confinement or diameter-size effects[26]. The PL energy shift induced by electron–hole separation is similar for each of the CdSe/CdS tetrapods, with the larger TP4 exhibiting only a slightly 'redder' emission compared with the others (Fig. 1e; 655 nm PL maximum versus ~645 nm in the case of TP1, TP2 and TP3). The differences in full-width-half-maxima (FWHM) of the ensemble PL peaks, where longer-arm tetrapods afford broader emission (~70 nm versus ~35 nm FWHM), can be attributed to a broader size distribution (~16–23% compared with 7–8% variation in tetrapod arm diameter in long- versus short-arm tetrapod populations, respectively), as well as intrinsic properties of the longer-arm tetrapods (see discussion pertaining to dual-red emission below).

The effect of tetrapod geometry is also apparent in PL decay-time trends. PL lifetimes were shown previously to increase with tetrapod arm length[1]. Comparing TP1 and TP3, we obtain a similar result, as decay times are clearly longer for the longer-arm TP3 (Fig. 1f). Furthermore, we observe that TP3 and TP4, which share similar arm lengths but differ in arm diameter, have similar PL lifetimes to each other. Taken together, these results suggest that the enhanced spatial separation between CdS-generated excitons and CdSe-core emissive states afforded by arm lengthening is the dominant factor increasing PL decay times. Interestingly, however, time-resolved PL results for short-arm TP2 tetrapods appear to conflict with this conclusion, as TP2 structures provide the longest PL decay times (Fig. 1f). In all cases, the decay curves are multiexponential and difficult to fit with a significant degree of confidence. Nevertheless, the marked increase in PL lifetimes observed for TP2 tetrapods suggests that shell/arm thickening and perhaps the shape change induced by SILAR growth have a significant impact on exciton decay dynamics. This may result from increased type II character or from reductions in contributions from fast non-radiative recombination processes, such as surface carrier trapping or Auger recombination (see discussion below).

**Shape-dependent blinking suppression.** An important photophysical property that is not accessible through characterizations of ensembles of nanocrystals is that of blinking. Ubiquitous in a range of fluorophores from molecular dyes and fluorescent proteins to QDs, nanorods and nanowires[27–32], blinking is important both fundamentally as a sign of 'things going wrong' in the exciton-to-photon conversion process and in its ability to impact applications from advanced bioimaging to display and lighting technologies[33–36]. It is now well known that shell thickness is the key variable influencing properties of CdSe/CdS

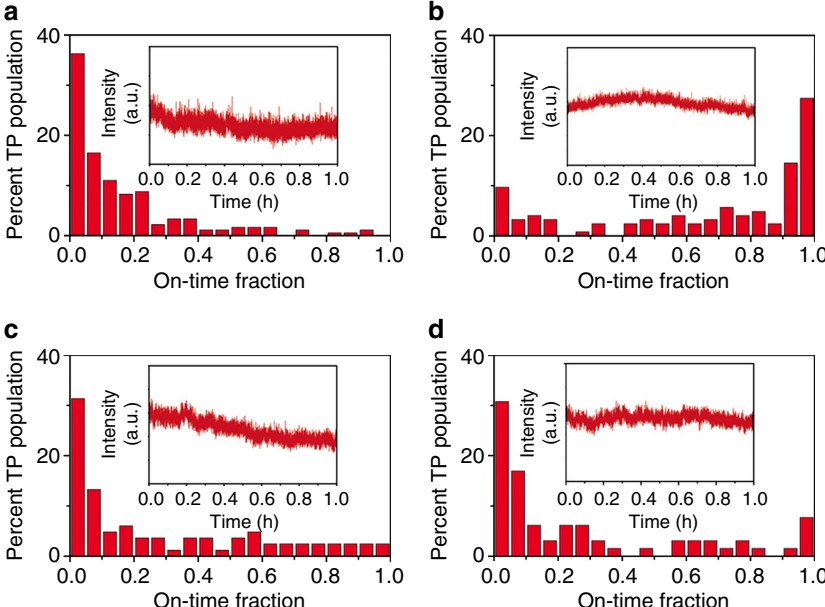

**Figure 2 | Dependence of on-time statistics and photobleaching on tetrapod shape from widefield microscopy.** Tetrapod (TP) on-time fraction histograms: (**a**) TP1 (thin/short arms), (**b**) TP2 (thick/short arms), (**c**) TP3 (thin/long arms), and (**d**) TP4 (thick/long arms). Insets show ensemble photobleaching behaviour for each type of tetrapod.

QDs[22–24,37]. Here we use the CdSe/CdS tetrapod as a platform for understanding how blinking is influenced by both thickening of the CdS shell and its transition from a zero-dimensional 'dot' into a one-dimensional rod.

To do so, we assessed time-dependent CdSe-core red PL behaviour of the four nanostructures at the single-tetrapod level using two different methods. The first—widefield microscopy—is the least forgiving, as single nanostructures are exposed to high power densities ($10\,W\,mm^{-2}$) under continuous-wave laser excitation for the duration of the experiment ($\sim1\,h$ here) and fluorescence is detected in widefield configuration using a charge-coupled device (CCD) camera, rather than a more sensitive single-photon counting approach (see Methods and Supplementary Note 2). The results of this experiment are shown in Fig. 2 in the form of on-time histograms. Fully non-blinking tetrapods would possess on-time fractions of 1.0 for 100% of the tetrapod population. Assessed in this way, we observe that only SILAR-overcoated tetrapods (TP2) are strongly blinking-suppressed over the 1 h observation time (Fig. 2). Fifty per cent of these tetrapods possess on-time fractions $>0.8$ and $\sim30\%$ are fully non-blinking. These results are similar to those obtained for unoptimized thick-shell CdSe/CdS QDs under similar experimental conditions[22]. Approximately 10% of TP3 and TP4 tetrapods possess on-time fractions $>80\%$, whereas for TP1 this fraction represents only $<2\%$ of the population. By comparison, conventional high-QY core/shell QDs are characterized by 0% of the population possessing on-time fractions $>80\%$, for example, CdSe/ZnS QDs[37]. Thus, each type of CdSe/CdS tetrapod nanostructure studied here is at least partially blinking suppressed and TP2 tetrapods are substantially blinking suppressed.

The widefield microscopy technique also provides information about permanent PL darkening or photobleaching. The average background-corrected fluorescence intensity of the individual tetrapods is plotted as a function of time (Fig. 2, insets). The resulting ensemble intensity versus time data show the relative tendency of each type of tetrapod to either remain emissive or to photobleach over the long 1 h interrogation period. By way of

comparison, we note that a typical photobleaching half-life for conventional core/shell QDs is $\sim15\,min$ (ref. 22). Here we find that PL intensity drops by only $\sim1/3$ of the initial intensity after $\sim0.5\,h$ for both short/thin (TP1) and long/thin (TP3) tetrapods, and the thick-arm tetrapods (TP2 and TP4) fare even better, resisting photobleaching entirely. The thick arms appear to act as thick shells do in the case of core/thick-shell QDs, for which a similar protective effect against fast photobleaching is now well known[37].

**Revealing the mechanism for CdSe/CdS tetrapod blinking.** Tetrapod blinking was further assessed using a more sensitive method—time-correlated single-photon counting employing a pulsed laser source and an avalanche photodiode (APD) detector (Supplementary Note 2). This approach affords important insight into the blinking mechanism. We find that although TP1 tetrapods exhibit on/off blinking behaviour, as suggested by wide-field microscopy measurements, blinking in the case of the other three tetrapods is better described as transitions between bright 'on' and dim or 'grey-state'[38] PL (Fig. 3), with TP2 again providing the fewest excursions into dark or dim-state emission. The latter can be seen in the histogram of blinking events (shown on the right-hand side of each blinking trace in Fig. 3).

Tracking PL intensity fluctuations simultaneously with PL lifetimes, in the form of fluorescence lifetime-intensity distribution diagrams (Fig. 3, extreme right of each panel, and Supplementary Figure 5 extended analysis of TP1 and TP3 correlated intensity-lifetime behaviour), shows that PL intensity changes are accompanied by changes in PL lifetimes. Such a correlation is characteristic of a charging/discharging blinking model[39]. Here, a tetrapod whose core is populated by an excess carrier (hole or electron) is 'charged' and as such is susceptible to either complete (TP1) or partial (TPs 2, 3 and 4) PL darkening by Auger recombination[40], which occurs with a simultaneous reduction in PL lifetime. Upon returning to an uncharged condition, the tetrapod emits at its full brightness with a longer PL lifetime. Significantly, the observation that TP2, TP3 and TP4 tetrapods can remain emissive in the charged condition shows

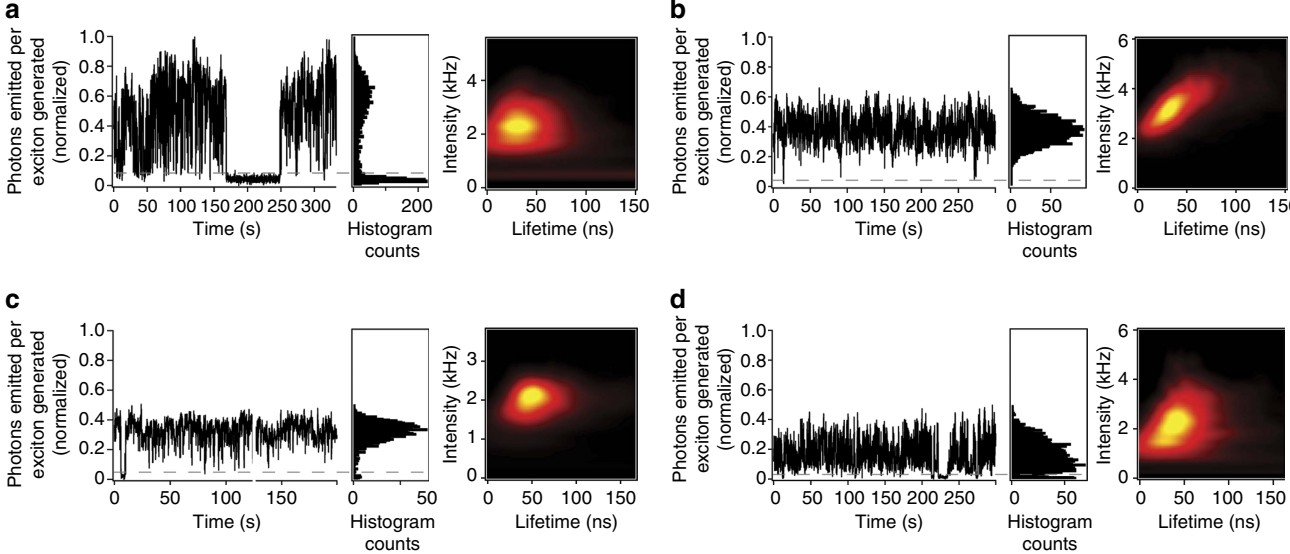

**Figure 3 | Single-photon counting experiments for different tetrapod shapes.** Representative intensity-time traces (normalized to photons emitted per exciton generated to facilitate comparisons between tetrapods) and fluorescence lifetime-intensity distribution (FLID) diagrams for single tetrapods reveal a range of blinking behaviours from on/off to on/grey-state fluctuations: (**a**) TP1 (thin/short arms), (**b**) TP2 (thick/short arms), (**c**) TP3 (thin/long arms) and (**d**) TP4 (thick/long arms).

that Auger recombination in the CdSe core is suppressed. In particular, the lowest intensity states of TP2 tetrapods are brighter than those of the other tetrapods (compare photons emitted per exciton generated in Fig. 3 for the lowest intensity emissions in each case; 'off' in the case of TP1 tetrapods). The relative brightness of TP2 grey-state emission suggests that Auger recombination is most suppressed in these tetrapods. This condition allows TP2 tetrapods to be functionally non-blinking, as observed in the widefield microscopy experiments.

**Evolution of single-tetrapod two-colour PL with pump fluence.** It has been proposed that suppression of Auger recombination is responsible for the existence of multiexcitonic red and green two-colour emission, with tetrapod volume (as achieved by arm lengthening) being the key controlling structural variable[16]. Here we clarify the role of tetrapod structure and distinguish between Auger processes active in the CdSe core and in the CdS arms, respectively. We obtain PL spectra for individual tetrapods for a range of pump fluences (405 nm pulsed laser excitation at 13, 38 and 116 $\mu$J cm$^{-2}$ per pulse; Fig. 4), corresponding in all cases to excitations of $\gg 1$ electron–hole pairs per tetrapod (Supplementary Fig. 1 and Supplementary Table 4). Importantly, by employing time-gated second-order autocorrelation function ($g^{(2)}$) analysis, we are able to confirm the single-nanostructure nature of the tetrapods being investigated (separating effects of possible tetrapod clustering from that of multiexciton emission on $g^{(2)}$ values; if $g^{(2)}$ values remain $> 0.5$ after a time-gating procedure is applied, then a cluster rather than a single nanostructure is being interrogated—see Supplementary Note 2)[41,42]. The single-tetrapod PL spectra show that with increasing pump fluence longer-arm tetrapods (TP3 and TP4) afford robust dual CdSe-red and CdS-green PL, while shorter-arm tetrapods (TP1 and TP2) do not.

However, the expected volume trends[16] for dual red and green emission are not strictly followed. We find that thick/short-arm variants (TP2) afford only weak green PL, even though these tetrapods possess larger volumes than thin/long (TP3) tetrapods (10,330 nm$^3$ compared with 6,840 nm$^3$) and similar volumes to thick/long TP4 tetrapods (10,330 nm$^3$ compared with 12,270 nm$^3$). Thus, arm length is the principal structural

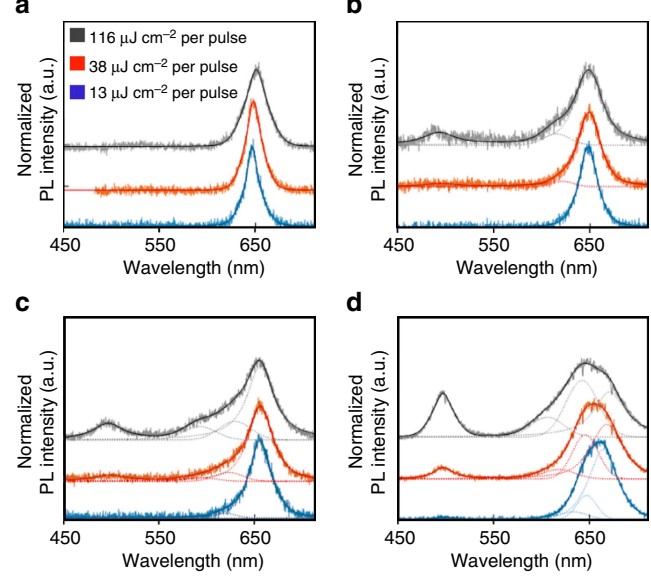

**Figure 4 | Emergence of two-colour emission as a function of tetrapod geometry and pump fluence in single-tetrapod spectra.** (**a**) TP1 (thin/short arms), (**b**) TP2 (thick/short arms), (**c**) TP3 (thin/long arms) and (**d**) TP4 (thick/long arms). Both raw and fitted data are shown. Smooth lines represent the best fit to the experimental data and dotted lines are the individual Gauss-Lorentzian peaks comprising the best fit to the experimental data. PL, photoluminescence.

variable controlling whether spatially defined two-colour emission occurs in seeded tetrapodal nanostructures and large tetrapod volume is not a sufficient condition for realizing this special case of two-colour emission. Notably, red and green dual emission in long-arm tetrapods is not rare, as 40% and 70% of TP3 and TP4 tetrapods, respectively, were found to be dual emitting in our single-particle analyses.

In addition to the emergence of green arm emission, the single-tetrapod spectra reveal the concurrent evolution of the red

portion of the spectra, including significant peak broadening and shoulder formation (Fig. 4). By fitting the spectra, two clear trends become apparent. First, shoulders now discernible as peaks are shifted to the blue from the primary CdSe core red emission by 50–200 meV (Supplementary Table 3). This range corresponds well to the low-temperature results obtained for emission from various multiexciton states in thick-shell CdSe/CdS QDs[43], for which it was shown that repulsive interactions cause biexciton emission to be shifted to higher energy by 13 meV, triexciton emission by 85 meV, higher-order multiexciton emission by 140–210 meV and charged-biexciton emission by 70–80 meV. Second, our results clearly show that green arm emission is observed only when multiexciton and charged-state core emission is also present, where the intensity of arm emission correlates with the integrated intensity of core multiexciton emissions (Supplementary Table 3).

**The mechanism for two-colour multiexcitonic emission.** Observations obtained from fitted pump-dependent single-tetrapod spectra are consistent with a state-filling model for green arm multiexciton emission. However, it remains unclear whether complete state filling, which according to previous reports would require occupation of the CdSe core by ∼30 electron–hole pairs[16], is required to reach a condition supportive of green arm emission. For this, a quantitative assessment is needed of the actual number of electron–hole pairs in the core during various stages of excitation. First, the number of excitons formed in the core and in the arms of each tetrapod was estimated based on excitation fluence and core and arm absorption cross-sections, respectively (Supplementary Fig. 1 and Supplementary Table 4). The calculation shows that direct excitation of the tetrapod cores results in formation of <1 electron–hole pairs even at the highest fluence for which single-tetrapod spectra were obtained, 116 μJ cm⁻² per pulse, whereas at this fluence almost 200 electron–hole pairs are generated in the CdS arms. A significant number of the carriers generated in the arms can migrate into the core either by direct relaxation into core band edge states or by exciton transfer or migration. However, the efficiency of the arm-to-core carrier/exciton transfer process, competing with carrier trapping and Auger recombination in the arms, is currently unknown. To determine the actual number of excitons contributing to red-core emission from either source (direct core excitation or transferred from an arm), we use transient absorption (TA) spectroscopy.

TA experiments were performed on TP2 tetrapods. TA spectra and the excited-state bleach relaxation dynamics at the bleach maximum for the CdSe core (640 nm) and the CdS arms (470 nm) as a function of pump fluence are shown in Fig. 5. Using the relaxation dynamics of the core bleach, normalized at long time delays (Fig. 5b, main panel), the average number of excitons per CdSe core at different excitation fluences can be determined from the ratio of the signal magnitude at the peak and at long delays where only the radiative decay is contributing to the relaxation (a/b ratio in Fig. 5b)[44]. Using this approach (Supplementary Note 4), we find that at the high fluence of 116 μJ cm⁻² per pulse, for example, the number of electron–hole pairs in the core is ∼2.5 on average. This is a small fraction of electron–hole pairs generated in the tetrapod as a whole at this pump fluence (∼200 electron–hole pairs, Supplementary Table 4). Thus, this result indicates that the vast majority (∼99%) of excitons generated in the arms do not reach the core.

Nevertheless, arm excitations do contribute to the observed core states. Based on CdSe core volume alone, only 0.63 electron–hole pairs on average should exist in the core

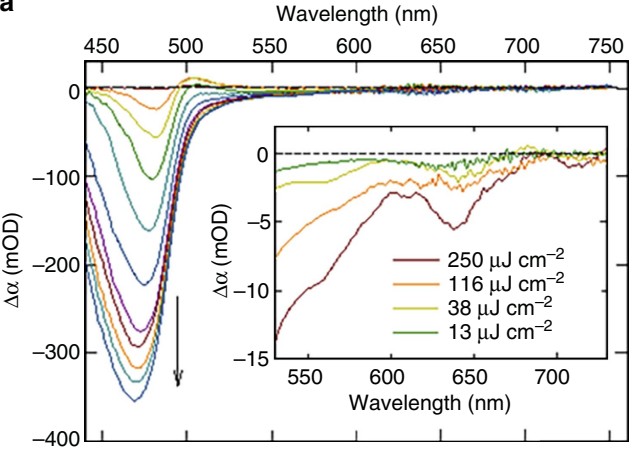

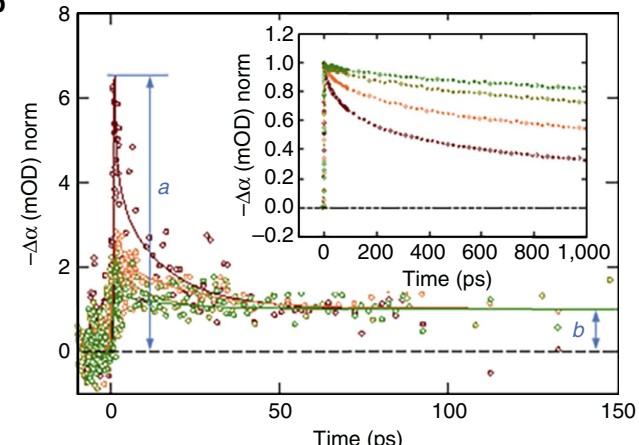

**Figure 5 | Fate of excited-state carriers generated in the CdS arms revealed. (a)** TA spectra of the thick/short-arm TP2 CdSe/CdS tetrapods in hexane recorded with $\lambda_{pump} = 400$ nm (3.1 eV) and pump fluence $j = 250$ μJ cm⁻² per pulse. The main panel shows the spectra recorded at pump-probe delay $\Delta t = 0$–2 ps, with ∼150 fs steps. The inset shows the dependence of the spectra on the pump fluence in the 500–750 nm range at $\Delta t = 2$ ps. **(b)** TA relaxation dynamics recorded at $\lambda_{mon} = 640$ nm at multiple pump fluences. The colour coding is consistent with the legend in the inset of **a**. The points are the experimental data and the solid lines are the best multiexponential fits to the experimental data. Both the experimental data and the fits were normalized at $\Delta t = 300$ ps. The inset shows the relaxation dynamics recorded at $\lambda_{mon} = 470$ nm, at multiple pump fluences. The kinetic traces were normalized at $\Delta t = 2$ ps.

(Supplementary Table 4), suggesting that the majority of core electron–hole pairs ($N_{e-h}$ ∼2 of ∼2.5) in the high-fluence, multiexciton regime have been generated in the arms. Significantly, even though only ∼1% of the electron–hole pairs generated in the arms reach the core, they push the core into the multiexciton regime and dictate the core relaxation dynamics. This result is consistent with the results of the PL analysis discussed above (Fig. 4), indicating that entry into the multiexciton regime in the core seems to be a pre-requirement for the observation of green-arm emission. Significantly, however, our results also indicate that complete core state-filling (on the order of 30 electron–hole pairs[16]) is not required to realize green arm emission.

A direct comparison of the TA bleach relaxation dynamics at the green (470 nm) and red (640 nm) wavelengths reveals that

there is no significant delay between the excited states located in the CdS arms and the CdSe core. This suggests that the CdSe core is not populated by a transfer or migration of the band-edge exciton from the CdS arm, but rather that the fraction of electron–hole pairs generated within the arms relaxes directly into the CdSe core during intraband cooling (that is, from high-energy arm excited states to band-edge core states). What is the fate of the rest of the carriers formed in the CdS arms? This question can be addressed by considering, for example, the results for low power excitation, that is, $13 \mu J \, cm^{-2}$ per pulse. At this pump fluence, $\sim 20$ excitons are generated on average in a tetrapod (see Supplementary Table 4) but no green PL is observed. According to TA data $< 1.5$ excitons on average relaxes through the core (Fig. 5b, main panel, green trace). This means that the majority of the generated excitons relax in the arms, as confirmed by the large CdS bleach observed in TA spectra at $\sim 470$ nm (Fig. 5a, main panel). However, no arm emission is observed in the single-tetrapod spectra at this pump fluence, which indicates that the arm excitons decay only through non-radiative channels.

To determine relative contributions from possible non-radiative channels—carrier trapping and Auger recombination—we consider both the appearance of the bleach signal and the bleach relaxation kinetics. First, the typical signatures of carrier traps (broad tail/band to the red of the main bleach) do not significantly contribute to the TA spectra (Fig. 5a, main). As shown previously[45,46], in the case of cadmium chalcogenides the TA signal is mostly sensitive to electron dynamics, rather than hole dynamics. Therefore, in further analysis we will assume that this observation pertains primarily to electron traps. Second, the TA bleach relaxation dynamics of the excitons in the arms (Fig. 5b, inset) for the $13 \mu J \, cm^{-2}$ per pulse data (green trace) show $\sim 20\%$ drop in signal intensity within 1 ns. This fast relaxation is attributed to the decay of carriers trough non-radiative channels, that is, trapping and/or Auger recombination. As even at this low fluence $\sim 20$ excitons are generated on average in the arms of a tetrapod, the Auger process probably contributes, at least in part, to the observed fast relaxation. This means that $< 20\%$ of the relaxation can be attributed to an electron trapping process. However, as no arm PL (proportional to the product of the number of electrons and holes) is observed at this fluence, this result implies that non-radiative relaxation is dominated by hole trapping, whereby band-edge electrons recombine non-radiatively with the trapped holes.

The above analysis indicates that the observation of dual red and green emission in CdS/CdSe tetrapods requires at least partial filling of hole traps in the arms and partial state-filling in the CdSe core. With the two relaxation channels partially or completely blocked (for example, through state filling or electrostatic repulsion) radiative decay in the arms becomes kinetically competitive and both red core emission and green arm emission are observed. The requirement of multi-level (but not complete) filling of the CdSe core states implies that the observation of dual emission is likely to be only when both the core and the arms are populated with multiple excitons. As in this regime radiative decay must compete with the non-radiative Auger process, the optimization of the dual PL requires effective suppression of Auger recombination in both the core and the arms of the tetrapod.

It has been shown that Auger recombination is strongly suppressed in thick-shell CdSe/CdS QDs[38,43,47], which are similar to the CdSe/CdS tetrapods in both their electronic structure (quasi type II band alignment) and physical structure (tetrapod arms effectively overcoat the CdSe core, aided further in the TP2 tetrapods by secondary shell/arm growth). We suggest that these structural similarities can account for the varying degrees of core Auger recombination suppression evidenced by both blinking statistics and evolution of multiexcitonic emissions (TP1 < TP3 ~ TP4 < TP2). With respect to arm Auger

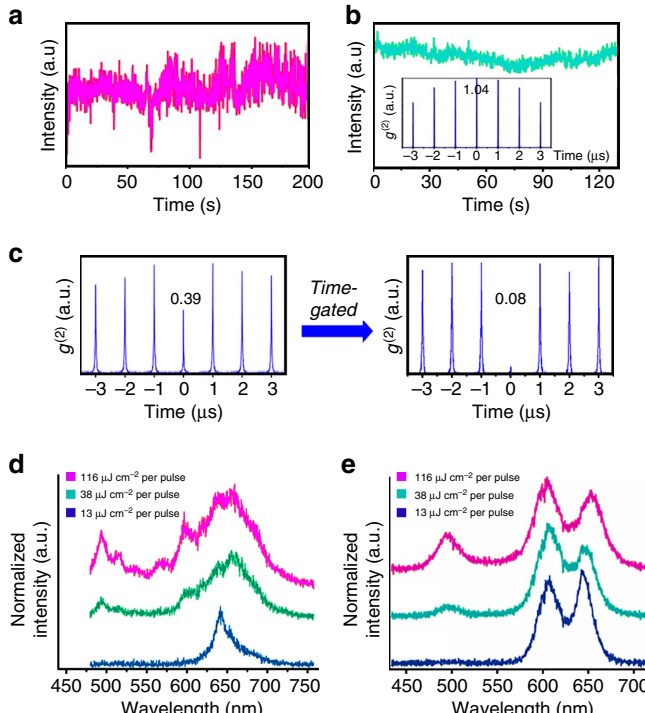

**Figure 6 | Blinking-suppressed multi-colour multiexcitonic emission from long-arm tetrapods.** (**a**) Single-tetrapod PL/time trace for red core emission. (**b**) Single-tetrapod PL/time trace for green CdS arm emission and accompanying $g^{(2)}$ trace (inset). (**c**) $g^{(2)}$ trace for red CdSe core emission before and after applying time-gating. (**d,e**) Pump fluence-dependent emission spectra. (**a–d**) Data obtained for same single TP4 tetrapod. (**e**) Data obtained for a single TP3 tetrapod.

recombination processes, prior work indicates that the transition from 0D (QD-like) to one-dimensional (1D; rod or wire-like) structure induces suppression of Auger-mediated decay of higher order multiexciton states ($> 2$ electron–hole pairs)[48]. For this reason, in the 1D-like tetrapod arm radiative recombination can effectively compete with non-radiative relaxation associated with the multiexcitonic states. Interestingly, the strongest green-arm emission is obtained from the longer-arm tetrapods, TP3 and TP4. As noted above, this observation is in contrast with the previously proposed simple volume dependence, and highlights the importance of arm length in reducing the efficiency of the arm Auger recombination process.

**Two-colour blinking suppression and tri-colour emission.** The practical effect of designing a nanostructure for both blinking suppression and two-colour emission is realizing the unique property of dual blinking-suppressed emission (Fig. 6). At sufficiently high pump fluences to generate multiple excitons in the arms and the core, some of the TP4 tetrapods, for example, exhibit both traits (Fig. 6a,b). PL intensity versus time trajectories taken at these fluences show steady red and green emissions. Time-gated $g^{(2)}$ values for the red core PL ($\ll 0.5$) prove that we are interrogating a single tetrapod (Fig. 6c). The approximately unity $g^{(2)}$ obtained for green-arm emission is unaffected by time-gating procedures (Fig. 6b, inset), as expected if the four tetrapod arms behave effectively as an assembly of four emitters rather than as a true single emitter.

Finally, CdSe/CdS tetrapods are not limited to dual emission. Some individual structures exhibit dual red PL (at $\sim 600$ nm and $\sim 650$ nm, respectively) at low pump power and, therefore, 'tri'

emission at high pump powers, with the addition of green PL (Fig. 6e). As we describe in the introduction, dual red PL has been attributed to the existence of two distinct processes: direct core emission and spatially indirect (quasi type II) core/shell emission resulting from the presence of a potential energy barrier at the core/arm interface (Supplementary Figure 4)[14,15]. We assign the peak at ~600 nm to the direct emission process (Supplementary Note 2)[15], which becomes more competitive with increasing pump power. We find that a longer arm length is needed to observe dual-red emission at room-temperature and speculate that the added strain induced by arm lengthening enhances the 'barrier' effect that underlies this process. Thus, a design criterion for increasing the probability of observing dual red PL at room temperature and, thereby, tri-emission at high pump fluence, is to further increase tetrapod arm length, with the caveat that simultaneous arm thickening would also be required to ensure optimal photobleaching and blinking characteristics.

In summary, using advanced seeded-growth and SILAR shell-growth techniques, we synthesized a tetrapod geometry series with which we showed that arm-diameter and arm-length engineering can be combined to control both multi-colour emission and photostability at the single-tetrapod level. The relative complexity of the asymmetric tetrapod heterostructure affords this added degree of control compared with non-blinking core/shell QD counterparts. By studying PL properties in single tetrapods, we showed that red and green emissions indeed derive from the same nanostructure and that dual emission is more commonly found in longer arm tetrapods compared to short-arm counterparts irrespective of arm thickness; the latter result implies that a large tetrapod volume is an insufficient condition for realizing dual emission. In addition, we showed that larger-diameter CdS arms, especially those thickened by SILAR shell growth, can afford significantly suppressed single-tetrapod blinking and photobleaching. We show that the mechanism for red CdSe core blinking in tetrapods is Auger-mediated charging/discharging intensity fluctuations. We also show that core emission transitions from purely excitonic to multi-excitonic/charged-state emission in the two-colour PL regime. To the extent that blinking is suppressed and that multiexcitonic emission is observed, Auger recombination in the core is necessarily suppressed. In addition, we describe the mechanism responsible for the observation of dual red-core and green-arm emission. Namely, processes of hole-trap filling in the CdS arms and partial state-filling in the core set the stage for two-colour PL, but partial suppression of Auger recombination in the arms is also required, which we attribute to the 1D character of the rod-like CdS arms. Thus, we demonstrate that the QD-seeded tetrapod is an ideal platform for exploring and tuning radiative and non-radiative excited-state processes for unprecedented blinking-suppressed multicolour multiexcitonic emission.

## Methods

**Synthesis.** Zinc-blende CdSe QDs (4 nm diameter) were synthesized according to a previously reported method and as described in detail in Supplementary Note 1 (ref. 17). These were used as 'seeds' for initiating CdSe/CdS tetrapod growth. If, instead, wurtzite CdSe QDs are employed as seed material, nanorods rather than tetrapods are produced[1]. Tetrapod synthesis followed the seeded growth approach of Fiore et al.[18], but with modified surfactant chemistry to reduce size dispersity in the resulting tetrapods[2,19,20]. In addition, ligand mixtures were used as the primary means for controlling tetrapod arm thickness and length. As shown in Supplementary Table 1, choice of phosphonic acid and ratio of phosphonic acid to oleic acid afforded control over tetrapod shape and access to TP1, TP3 and TP4 tetrapods. Thick-but-short tetrapods (TP2) were synthesized from TP1 tetrapods by employing SILAR shell growth to increase the thickness of the arms (shell) by ~5 CdS monolayers[21]. As described previously for the synthesis of non-blinking CdSe/CdS core/thick-shell QDs[24], long anneal times were employed following each successive addition of Cd and S precursors (2 and 1 h, respectively) to improve single-particle optical properties.

**Widefield microscopy to assess blinking and photobleaching.** Ultra-dilute tetrapod solutions were prepared and in all cases immediately drop-cast on glass substrates to achieve tetrapod densities of ~0.02 tetrapod per μm². Samples were imaged under continuous wave 405 nm laser excitation in a home-built wide-field (40 × 40 μm) optical microscope. PL signal was recorded over 1 h using liquid N₂-cooled CCD detector with 0.1 s integration time per frame. Average fluorescence intensity of the individual tetrapods versus time data represents the relative tendency of each type of tetrapod to either remain emissive or to photobleach over the long 1 h interrogation period (Fig. 2, insets). Individual tetrapods were also assessed for their respective tendency to produce emission above the threshold value as a function of time. The resulting blinking behaviour for individual tetrapods was used to create on-time fraction histograms (Fig. 2).

**Single-photon counting experiments.** To measure $g^{(2)}$ values and lifetimes of single tetrapods, a 405 nm pulsed laser (~80 ps pulse width) and a laser scanning confocal microscope equipped with a 100 ×, 0.85 numerical aperture objective were used to excite the tetrapods. PL from single tetrapods was collected by the same objective lens and then directed to two APDs forming a Hanbury–Brown–Twiss spectrometer. This allowed for the identification of single nanostructures via determination of single-photon emission behaviour of the low-energy emission under low excitation power density ($\{N\} \ll 1$). For measurements of single tetrapod lifetime, $g^{(2)}$, and most spectra, the repetition rate was 1 MHz. For the high-power single tetrapod spectra measurement, the repetition rate was chosen to be 3 MHz, to enhance excitation of multiexcitons. To identify dual-emitting tetrapods, switchable 510 nm short-pass filters and 610 nm long-pass filters were installed in front of the two APDs. With different filters before the two APDs, a quick dual-channel scan identifies emissive spots that radiate strongly at both short band and long band simultaneously.

To obtain increasing fluences of 13, 38 and 116 μJ cm⁻² per pulse for power-dependence measurements, the same single tetrapod was investigated at each power, beginning with the lowest power. The laser power was then adjusted to increase the measured power at the sample for each subsequent measurement. Low-frequency excitation was performed at 1 MHz, whereas high-frequency excitation was performed at 3 MHz. Spectra were recorded by dispersing the collected single tetrapod emission onto a CCD array using a Princeton Instruments Acton SP2300 Spectrometer. Single-tetrapod data were collected by averaging three spectra that were integrated for 30 s each.

**TA spectroscopy.** The TA spectra were recorded using a commercial pump–probe TA spectrometer (Helios, Ultrafast Systems). The femtosecond pump pulses at energy of 3.1 eV (400 nm) were generated by focusing a portion of the 800 nm output from 1 kHz regenerative amplifier (Spectra Physics) onto a BBO crystal. The probe pulse, a white light continuum, was generated by passing the second portion of the 800 nm output through a delay line and a 2 mm sapphire crystal. The instrument response time of the system is 220 fs (FWHM). The diameter of the probe beam focused onto the sample was ~0.1 mm. The pump fluence was varied from 13 to 250 μJ cm⁻² per pulse. To minimize the potential effects of photo-degradation and local heating, the sample solution was continuously stirred during the experiment.

**Data availability.** The data sets generated during the current study are available from the corresponding authors on reasonable request.

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

## Acknowledgements

This study was supported primarily by a Division of Materials Science and Engineering, Office of Basic Energy Sciences (OBES), Office of Science, U.S. Department of Energy (DOE) grant 2009LANL1096. Work of UT Dallas group (S.K. and A.V.M.) was supported by a DOE OBES grant DE-SC0010697. Work was performed at CINT, a DOE, OBES Nanoscale Science Research Center & User Facility with aspects of the work supported by CINT User Projects (U2013A0134 and U2013B0037).

## Author contributions

N.M. conducted all syntheses, under the guidance of J.A.H. N.J.O., F.W. and Z.H. conducted single-tetrapod optical properties and time-resolved measurements, under the guidance of H.H. S.K. conducted ensemble absorption and PL properties measurements, under the guidance of A.V.M. and J.A.H. M.S. conducted the TA experiments and analysed the ensemble optical spectroscopy data. J.L.C. performed electron microscopy analyses. J.A.H. wrote the manuscript with key contributions from M.S., N.M. and all other authors.

## Additional information

**Competing interests:** The authors declare no competing financial interests.

**Publisher's note**: 

