## [Peer Review File · Nature Communications]

Reviewers' comments:

Reviewer #1 (Remarks to the Author):

This submission by Hollingsworth and co-workers is very interesting and at the forefront of nanoscience research. What is particularly impressive is that the authors have taken a complex heterostructure tetrapod system and tuned the physical properties of arm length, arm diameter and composition to control the emission achieving dual and tri-band emission over two distinct colours. The authors have carried out a comprehensive study that goes substantially beyond what has been reported to-date and in one report have set a bench-mark for the possibilities for these types of nanostructures. Particularly impressive is that the authors have reliably produced 4 distinct tetrapod morphologies and have determined the shape and size effects that give rise to key optical properties such as PL lifetimes, PL blinking and photobleaching in addition to the dual emission characteristics. I would recommend publication with minor revisions.

The work is clearly of fundamental importance and the authors state that the structures and dual emission properties are of technological significance although it is not stated clearly where the potential application set is. The authors should comment on this in the introduction and in the conclusion section of the manuscript include some perspective on potential further developments.

A comment is that tetrapods are known to flatten out when deposited on a substrate due to capillary forces, have the authors considered the impact of this strain on their PL measurements. It is likely that this would be more apparent for longer arm tetrapods and would not be evident in the shorter arm analogues and may have an impact on their optical properties.

In page 18 the observation of dual read PL is reported and is attributed as being similar to two previous works. This paragraph is written on the basis that the reader has a deep knowledge of these works and it would be better if a discussion of why this PL occurs in the context of the authors samples and mechanism involved and state that this is in agreement with the previous works. Furthermore Pg 18, 'Ref.15' is written in the text, this should be described as the first or corresponding author name or just indicated as a normal reference

Reviewer #2 (Remarks to the Author):

This work demonstrates the PL characteristics at the level of single tetrapod and firstly investigates the multiexcitonic emission in individual nanostructures. The tetrapod arm length and thickness dependent blinking, photo bleaching are studied, which could facilitate to distinguish effects of tetrapod arm length from those of particle volume and reveal the suppression of Auger recombination is not the sufficient condition to realize dual multiexcitonic emission. All of these results would be helpful to understand the effects of geometry engineering on PL photo-stability and complexity in asymmetric core/shell structures.

The paper is well-organized and reasonable, with appropriate references to related experimental and theoretical work. Given the great interest in this system and that this paper presents comprehensive investigations, I recommend that this paper be accepted for publication. However, the following points should be revised.

(1) Some words are written down with italic in the last paragraph, for example, the word "single", "both" in the last paragraph in Page 4, any meaning on that?

(2) In Figure 1, only Figure 1(a) shows the scale bar. But how about Figure 1(b), (c), and (d)? Moreover, in Figure 1(e), the PLE spectra are normalized to the same intensity at 525nm. But honestly speaking, the peak around 525nm really cannot be readable. I suggest here there should be an enlarged figure for this wavelength range.

(3) In Figure 3(d), with the per pulse excitation power at 11 μ W, the emission peak at the low energy side should be fitted as it is not a single peak. Please explain why additional peaks appear.

Reviewer #3 (Remarks to the Author):

The manuscript by Mishra et al. reports on the synthesis of nanotetrapods with a CdSe core and CdS arms. The synthesis is targeted to tetrapods of different arm length in thickness to explore at the ensemble and single particle level the influence of arm length and overall particle volume on PL emission at different wavelength. The main conclusion of the manuscript is that particle volume alone is not a sufficient condition to achieve dual emission at CdSe core and CdS arms, but length of CdS arms is an important aspect.

Novelty. The work is an extension of what has been done already in ensemble (reference 16) and at the single particle level (ref. 14 and 15). Although it is based on a well planned series of systematic structures and using several techniques it does not reach results that are particularly original or new with respect to what is known already. It is a complementary work that could find dissemination in journals such as Scientific Reports, JPC-C or Nanoscale, but certainly not in Nature Communications.

The experimental methodology is of high quality with state of the art synthesis techniques and also advanced experiments such as photon anti-bunching on single particles. What is probably lacking to achieve an excellent work is theoretical support, for example showing the role of wavefunction overlap in minimizing Auger recombination. In general, an aspect on which the authors should have worked more extensively is the evaluation of the number of excitons per nanoparticle in their experiments. Absorption cross section are known for these nanomaterials, which are actually out in the community since almost a decade (ref.1). There is a good statistics (number of particles studied) for blinking experiments, but the statistics on the TEM measurements is instead not mentioned, on how many particles the measurements of size have been performed and what are the errors in the calculation of the volumes? Another aspect which I find well below the standards for publication in Nature Comm is the analysis of the ensemble data, what is the rationale for the normalization of the PLE spectra? This seems to be entirely arbitrary... what are the concentration of the solutions or films (not clear from the SI if these PLE measurements are performed in solution or films, while for time resolved it is clear that has been done on films).

There is some speculation on the role of tapered arms and cylindrical. This is, however, not substantiated with a quantitative discussion. What is the exciton Bohr radius for bulk CdS? How the authors expect the electron and hole wave functions are spread in the CdS upon excitation at 405 nm?

The presentation could be improved. The manuscript is very long and I believe it is too long because the authors are commenting too much on results from previous papers out in the literature. The flow is often interrupted with comparisons with giant dots that could be group in a final section discussing the results in a comprehensive picture. While it is too long and with far too many references to only partially related work, it misses the explanation of those concepts that are crucial for a broad readership to understand. Auger recombination and multi-excitons states, which are clearly two of the most important aspects of this work are not introduced properly with an explanation of what they are and how important they are for the photophysics of semiconductor nanocrystals.

I cannot recommend this work for Nature Communications.

We are grateful to all of the reviewers for their helpful suggestions and believe that the revised manuscript has been considerably improved through this process.

Summary of novelty, significance, and advances over previous work:

- Benchmark report for room-temperature tetrapod blinking properties
 - First assessment of blinking/bleaching statistics and trends (**substantially enhanced in revision by addition of complementary experiments/analyses** – see summary of changes below).
 - First report defining the blinking mechanism for tetrapod nanostructures (**new to revision** – see summary of changes below).
 - First description of using structure-engineering to suppress blinking in tetrapod nanostructures.
 - Culminating in first ever multi-color, multi-excitonic suppressed blinking PL.
- First study of multiexcitonic dual emission at the single-nanostructure level.
- New mechanistic insight into the property of multi-excitonic two-color emission
 - First experimental investigation of two-color multiexcitonic emission in quantum-dot seeded tetrapods (represented here by the CdSe/CdS system) sufficiently complete and quantitative to enable a definitive assessment of the unique multiexcitonic dual-emission mechanism (**new to revision – substantially improved data analysis per reviewer suggestions and addition of highly informative transient absorption spectroscopy experiments** – see summary of changes and detailed responses to reviewers).
 - Using a systematic and comprehensive structure series (accessing both tetrapod arm diameter and arm length effects, in contrast with literature report) and combining numerically aided analysis of single-tetrapod fluence-dependent PL data (**first of its kind; analysis new to revision**) with ultrafast pump-probe (transient absorption) experiments (**new to revision**), we provide important clarification and significant new insight regarding the roles of nanostructure, hole traps and core state-filling in realizing multi-excitonic two-color emission.
- “Design principles” established for engineering asymmetric nano-heterostructures for novel functionality of blinking-suppressed multiexcitonic (electronically coupled) two-color photoluminescence.

Summary of changes:

- Modification of Figure 1 to address reviewer comments pertaining to photoluminescence excitation (PLE) data and transfer of PLE results to SI, along with addition of new PLE discussion in SI.
- Thorough assessment of shape-dependent blinking properties – including blinking suppression and the blinking mechanism – for CdSe/CdS tetrapods. We now use two methods to assess tetrapod blinking: widefield microscopy (previous and current Figure 2) and time-correlated single-photon counting (TCSPC) (**new Figure 3 and Figure S5**) that afford rigorous assessment of blinking/bleaching statistics (previous) and insight into the blinking mechanism (**new**), respectively. (*Novelty/significance enhanced*)
- Numerical fitting of single-tetrapod photoluminescence (PL) spectra (**new Figure 4 and new Table S3**) and resulting new insight into correlations between two-color PL evolution and multiexcitonic/charged-emission processes in the CdSe core. (*Reviewer request addressed.*)
- Quantitative assessment of tetrapod absorption cross-sections and number of electron-hole (e-h) pairs generated in the tetrapods as a function of pump fluence in both the arms and the core (**new main text discussion, new SI calculation/method, new Table S4 and new Figure S1**). (*Reviewer request addressed.*)
- New transient absorption (TA) experiment and analysis: Quantitative assessment of arm-to-core “antenna” effect to answer, “how many excitons created in the arms reach the core?” (**new Figure 5 and main text discussion and Method**).
- New transient absorption (TA) experiment and analysis: Achieved new insight into the mechanism responsible for two-color multiexcitonic emission in seeded tetrapods (by **advanced analysis of data presented in new Figure 5, as explained in new main text**). (*Reviewer request for better support of mechanistic claims addressed.*)
- More careful and deeper analysis of original and new ensemble and single-tetrapod data. (We note that TEM statistics were previously provided.)
- Significant editing of the text to improve presentation.

Detailed responses to all reviewer comments

Reviewer #1 (Remarks to the Author):

Comment: This submission by Hollingsworth and co-workers is very interesting and at the forefront of

nanoscience research. What is particularly impressive is that the authors have taken a complex heterostructure tetrapod system and tuned the physical properties of arm length, arm diameter and composition to control the emission achieving dual and tri-band emission over two distinct colours. The authors have carried out a comprehensive study that goes substantially beyond what has been reported to-date and in one report have set a bench-mark for the possibilities for these types of nanostructures. Particularly impressive is that the authors have reliably produced 4 distinct tetrapod morphologies and have determined the shape and size effects that give rise to key optical properties such as PL lifetimes, PL blinking and photobleaching in addition to the dual emission characteristics. I would recommend publication with minor revisions.

Response: We thank the reviewer for his/her highly positive assessment of the quality of our work, as well as its novelty (“...comprehensive study that goes substantially beyond what has been reported to-date...have set a bench-mark for the possibilities for these types of nanostructures.”)

Comment: The work is clearly of fundamental importance and the authors state that the structures and dual emission properties are of technological significance although it is not stated clearly where the potential application set is. The authors should comment on this in the introduction and in the conclusion section of the manuscript include some perspective on potential further developments.

Response: We apologize for any lack of clarity on our part. In the first paragraph of the main text of the manuscript we provide two examples of technological applications enabled by this class of nanomaterial, namely, solar cells and lasing. These pertain to basic ensemble-level properties of core/arm tetrapods, e.g., absorption cross-section and long biexciton lifetimes. With respect to dual-emission applications, we state in the second paragraph, “Dual-emitting nanostructures are... technologically significant as sources for ratiometric temperature sensing^{4,5} or potentially white-light generation.⁶” This statement was intended to apply to all classes of dual-emitting nanostructures. We have modified the text in these sections for enhanced clarity.

Comment: A comment is that tetrapods are known to flatten out when deposited on a substrate due to capillary forces, have the authors considered the impact of this strain on their PL measurements. It is likely that this would be more apparent for longer arm tetrapods and would not be evident in the shorter arm analogues and may have an impact on their optical properties.

Response: It is indeed possible that enhanced strain in the solid-state could impact PL properties of our single-tetrapod emitters. However, we note that our observations of enhanced green emission in longer-arm tetrapods are at least in qualitative agreement with that obtained for ensembles of tetrapods measured in solution, i.e., we are not simply reporting an effect resulting from strain. Furthermore, we find that our solution-phase ensemble PL spectra are also qualitatively similar to our solid-state single-tetrapod PL spectra in that the shorter-arm tetrapods in both cases yield narrower-band spectra compared to their longer arm counterparts. Based on prior work that subjected tetrapods to pressure-induced arm bending (Ref. 3) one might be tempted to conclude that the broader PL associated with longer-arm tetrapods is a result of strain, but it is present in our case in both solution-phase/ensemble and single-tetrapod PL spectra, perhaps suggesting that arm-strain is not fully eliminated in solution-suspended tetrapods, or that there is a different or, at least, less simple, origin for the broader emission. We note, for example, in the manuscript that the existence of “dual-red” emission is attributable to enhanced strain in longer-arm tetrapods and may contribute to broadened red emission in this system. Taken together, arm-length-derived strain effects are present and potentially exaggerated in the solid-state, but these do not fundamentally alter the trajectory of multiexcitonic green arm-emission processes that are the focus of our study.

Comment: In page 18 the observation of dual red PL is reported and is attributed as being similar to two previous works. This paragraph is written on the basis that the reader has a deep knowledge of these works and it would be better if a discussion of why this PL occurs in the context of the authors samples and mechanism involved and state that this is in agreement with the previous works. Furthermore Pg 18, 'Ref.15' is written in the text, this should be described as the first or corresponding author name or just indicated as a normal reference.

Response: Again, we apologize for any lack of clarity on our part. We have now modified the text on pages 22 and 23 (formerly page 18) to refer the reader back to our introduction to dual-red emission (and its associated mechanism) in paragraph 3 (pages 3-4). On page 22-23 the modified sentences read, “Some individual structures exhibit dual red PL (at ~600 nm and ~650 nm, respectively) at low pump power and, therefore, “tri” emission at high pump powers, with the addition of green PL (Figure 6e). As we describe in the introduction, dual red PL has been attributed to the dual existence of direct core emission and spatially indirect (quasi type II) core/shell emission resulting from the presence of a potential energy barrier at the core/arm interface (Figure S4).^{14,15} The language used to refer to the references is no longer an issue in the modified text.

Reviewer #2 (Remarks to the Author):

Comment: This work demonstrates the PL characteristics at the level of single tetrapod and firstly investigates the multiexcitonic emission in individual nanostructures. The tetrapod arm length and thickness dependent blinking, photo bleaching are studied, which could facilitate to distinguish effects of tetrapod arm length from those of particle volume and reveal the suppression of Auger recombination is not the sufficient condition to realize dual multiexcitonic emission. All of these results would be helpful to understand the effects of geometry engineering on PL photo-stability and complexity in asymmetric core/shell structures.

The paper is well-organized and reasonable, with appropriate references to related experimental and theoretical work. Given the great interest in this system and that this paper presents comprehensive investigations, I recommend that this paper be accepted for publication. However, the following points should be revised.

Response: We thank the reviewer for his/her highly positive assessment of the quality of our work, especially with respect to the significance of this materials system and our contribution to a new comprehensive understanding of “*multiexcitonic emission in individual nanostructures*” allowing us for the first time to “*distinguish effects of tetrapod arm length from those of particle volume and reveal the suppression of Auger recombination is not the sufficient condition to realize dual multiexcitonic emission.*”

Comment: (1) Some words are written down with italic in the last paragraph, for example, the word "single", "both" in the last paragraph in Page 4, any meaning on that?

Response: In both cases, we aim to emphasize the unique aspects of our work. By italicizing “single” we emphasize that we study multiexciton processes at the level of single tetrapods, while by italicizing “both” we emphasize that our study includes arm thickness as a structural variable in addition to arm length. We suggest that the Editor determine whether use of italics is appropriate in these cases. That said, in the revised manuscript we have changed *single* to single.

Comment: (2) In Figure 1, only Figure 1(a) shows the scale bar. But how about Figure 1(b), (c), and (d)?

Response: We have added to Figure 1 caption the explanation: “[Scale bar in (a) applies also to (b)-(d).]” We thank the reviewer for pointing out this deficiency in our text.

Comment: Moreover, in Figure 1(e), the PLE spectra are normalized to the same intensity at 525nm. But honestly speaking, the peak around 525nm really cannot be readable. I suggest here there should be an enlarged figure for this wavelength range.

Response: There is no “peak” at this wavelength. Our only intention is to normalize at a wavelength (energy) that is below the bandgap of the CdS arm, i.e., at a wavelength for which PLE intensity is only responsive to the CdSe core. Due to ensemble effects and, more importantly, softening of absorption and PLE features resulting from electronic wavefunction spreading (quasi type II core/shell band alignment paired with large-volume structure), clear peaks are no longer present in the spectra, even for wavelengths only related to the CdSe core. We believe that normalizing at this CdSe core wavelength allows us to assess relative contributions to PLE of the CdS arms at lower wavelengths (higher energies), as the spectra are normalized for CdSe core contributions, where the CdSe component for each tetrapod is identical.

Nevertheless, we understand that normalizing in a region characterized by only weak PLE likely introduces uncertainty to our PLE comparisons. For this reason, we have decided not to highlight the PLE data in the main text. We still include it in the SI (Figure S6) as a *qualitative* assessment of relative arm-to-core “antenna” effects for the different tetrapod structures. For a quantitative determination of the actual number of electron-hole (e-h) pairs transferred from the arms to the core, we now use transient absorption (TA) spectroscopy combined with calculations of e-h pair generation based on absorption cross-sections and fluence (see below extensive discussion in response to comments by Reviewer 3). Unfortunately, however, unlike PLE experiments, absorption measurements are hindered in the case of significant scattering. Only TP2 tetrapods afforded sufficiently clear (non-scattering) suspensions to support high quality TA analysis. Also for this reason, we have decided to keep the PLE data as part of the SI, as it provides at least qualitative insight into the relative extent to which arm excitations reach the core in each of the tetrapods, revealing that arm volume alone does not predict the extent to which arm excitation contributes to core emission.

Comment: (3) In Figure 3(d), with the per pulse excitation power at 11 μ W, the emission peak at the low energy side should be fitted as it is not a single peak. Please explain why additional peaks appear.

Response: We thank the reviewer for his/her suggestion, and we have now fit all spectra in Figure 3, not only 3d – Figure 4 in the revised manuscript. **Table S3** has been added to Supplementary Information, showing the numerical results for peak fitting. Compared to low-temperature experiments, the peaks obtained by our room-temperature experiment are broader, and there is more overlap, rendering peak fitting less able to define distinct peaks in every case where they may exist. *Nevertheless, two clear trends are apparent.* First, shoulders discernible as peaks through fitting are shifted to the blue from the primary CdSe core red emission by 50 to 200 meV (**new Figure 4 and Table S3**). This range corresponds well to the low-temperature results obtained for emission from various multiexciton states in thick-shell CdSe/CdS QDs (new reference 44), for which it was shown that repulsive interactions cause biexciton emission to be shifted to higher energy by 13 meV, triexciton emission by 85 meV, higher order multiexciton emission by 140-210 meV, and charged-biexciton emission by 70-80 meV. Second, our results clearly show that green arm emission is observed only when multiexciton core emission is also present, where the intensity of arm emission correlates with the integrated intensity of core multiexciton emissions (**new Table S3 I_{rel} values**). In this way, the observations obtained from fitted pump-dependent single-tetrapod spectra are consistent with a state-filling model for green arm multiexciton emission. However, they do not provide an answer to a more subtle but important question: “what is the extent of state-filling required to reach a condition supportive of green arm emission.” Literature implies that complete core state filling is required (~30 electron-hole pairs) to reach the condition supportive of green-arm emission (Ref. 16). We now provide a quantitative assessment of the actual number of excitons in the core as a function of pump fluence, *as well as an assignment of their origin (core or arms) – see response to Reviewer 3 comments below.*

Reviewer #3 (Remarks to the Author):

Comment: The manuscript by Mishra et al. reports on the synthesis of nanotetrapods with a CdSe core and CdS arms. The synthesis is targeted to tetrapods of different arm length in thickness to explore at the ensemble and single particle level the influence of arm length and overall particle volume on PL emission at different wavelength. The main conclusion of the manuscript is that particle volume alone is not a sufficient condition to achieve dual emission at CdSe core and CdS arms, but length of CdS arms is an important aspect.

Response: In addition to assessing the tetrapod-geometry requirements for efficient two-color emission, distinguishing between volume and length effects as the reviewer indicates, we also provide important new insight into tetrapod blinking behavior, and now the blinking mechanism and the actual mechanism of dual emission. Finally, we define the nanostructure “engineering” requirements for realizing novel two-color multiexcitonic blinking suppression. Thus, we suggest that our manuscript provides several new and important conclusions pertaining to the unusual properties accessible via the seeded-tetrapod geometry. To clarify and deepen our conclusions, we provide in the revision further elaboration on key topics – the blinking mechanism, a quantitative description and mechanism for two-color multiexcitonic emission, and the relation of Auger recombination processes in the core and the arm to blinking and dual emission, respectively. These changes and additions to the manuscript are summarized in the introduction to this response and are described in detail in our responses to specific comments.

Comment: Novelty. The work is an extension of what has been done already in ensemble (reference 16) and at the single particle level (ref. 14 and 15). Although it is based on a well planned series of systematic structures and using several techniques it does not reach results that are particularly original or new with respect to what is known already. It is a complementary work that could find dissemination in journals such as Scientific Reports, JPC-C or Nanoscale, but certainly not in Nature Communications.

Response: We respectfully disagree with the reviewer on this point. We in fact study for the first time multiexcitonic two-color (red and green) emission at the level of single tetrapods. Refs. 14 and 15 only consider red emission associated with the CdSe core. The “two colors” in those works entail only excitonic core emissions, slightly offset from one another due to differences in energy between type I (conduction band electron confined to core) and quasi type II (conduction band electron delocalized into shell) band alignments. *These are not multiexciton processes.* We, too, observe two red emissions in some tetrapods, but this topic is not the focus of our work.

Ref. 16 does indeed describe the multiexciton process of dual red-core and green-arm emission; however, as we point out in our manuscript this work is limited to ensemble measurements. In contrast, by studying these tetrapods at the single-tetrapod level we are able for the first time to definitively show that the observed red and green emissions derive from single nanostructures. Moreover, we have – also for the first time – elucidated the blinking properties of core/arm tetrapods. In this way, our study provides the benchmark for core/arm tetrapod blinking statistics and photobleaching trends (Figure 2). We even demonstrate that blinking can be suppressed at the very high pump fluences required to induce multiexcitonic green arm emission, *revealing the unique property*

of suppressed-blinking two-color multiexcitonic emission (**revised as new Figure 6**). And significantly, we do not simply study blinking, we also show how tetrapod structure can be engineered to achieve strong blinking suppression, which itself remains an elusive property for nanoscale semiconducting materials in general and is an area of active research.

To further strengthen the impact of our work in the context of tetrapod blinking, we have added results (**new Figure 3 and Figure S5**) that reveal for the first time the mechanism underlying tetrapod blinking events. Specifically, using a more sensitive time-correlated single-photon-counting (TCSPC) method for assessing time-dependent single-tetrapod PL, we show that apparently on/off blinking events in TP3 and TP4 tetrapods are better characterized as on/"grey" or dim-state events, as these tetrapods rarely actually turn off. By analyzing the relationship between PL intensity and PL lifetime in single tetrapods (using fluorescence-lifetime-intensity-distribution, or FLID, diagrams – also in new Figure 3), we show that blinking events follow the charging/discharging model that describes "grey" or dim-emission events as times during which the nanocrystal is charged, and PL (radiative recombination of the electron-hole pair, or exciton) is suppressed as a result of efficient non-radiative Auger-mediated recombination. *Thus, we can now conclusively correlate blinking in this system with Auger processes.*

TP4 tetrapods – the "non-blinking" tetrapods based on our widefield microscopy blinking experiments (Figure 2) – are also found to fluctuate between bright and dimmer states. Importantly, however, the lowest intensity states of TP2 tetrapods are brighter than those of the other tetrapods. Here, we compare **photons emitted per exciton generated** in the **new Figure 3** for the lowest intensity emissions in each case. The relative brightness of TP2 grey-state emission allows these tetrapods to be functionally non-blinking, as observed in the widefield microscopy experiments (Figure 2), and suggests that Auger recombination in the CdSe core (as the blinking measurements study red core emission) is most suppressed in these tetrapods, compared to TP1, 3 and 4. In contrast, we find that TP1 tetrapods do experience true on/off blinking behavior. Taken together with our new analysis of multiexcitonic emission processes in the CdSe core (revised single-tetrapod PL spectra, **new Figure 4**), we are now able to more clearly assign relative degrees of Auger suppression in the four tetrapods of our structural series – TP1<TP3,TP4<TP2. These important clarifications/distinctions are now made in the revised text (**pages 12 to top of 14, and page 21**).

*In addition, as summarized above and described below, we now present for the first time a quantitative and definitive mechanism for two-color multiexcitonic emission, as supported by our now more fully elaborated experimental study.

Comment: The experimental methodology is of high quality with state of the art synthesis techniques and also advanced experiments such as photon anti-bunching on single particles. What is probably lacking to achieve an excellent work is theoretical support, for example showing the role of wavefunction overlap in minimizing Auger recombination.

Response: We respectfully disagree with the reviewer on this point. We believe that the conclusions we are able to make based on experimental investigations – *now bolstered with new confocal TCSPC blinking data, numerical analysis of single-tetrapod PL spectra, transient absorption (TA) spectroscopy and advanced analysis, and PL saturation data and related calculations of excitation levels* – are substantially new, sufficiently well supported and would not be significantly enhanced by addition of theoretical modeling. In fact, in addition to defining the mechanism for blinking behavior, we now provide a non-speculative and well substantiated mechanism for the dual emission process itself (see below).

A theoretical depiction of electron-hole wavefunction overlap already exists in the literature for CdSe/CdS tetrapods (Ref. 16). With this, the authors were simply able to show the result of the well-known quasi type-II band structure on electron-hole overlap in the CdSe core, i.e., the electronic wavefunction spreads into the shell (arms), while hole states are confined to the core. This picture has also been developed previously for CdSe/CdS QDs and nanorods. Ref. 16 further suggests that arm electron and hole states can co-exist with the core states following state-filling in the core. In that case, the pictorial depiction obtained from modeling (though no numerical results were provided in their paper) shows electron and hole states occupying identical locations in the arms. This theoretical result does not afford substantial insight into the structural dependency of arm Auger recombination processes; rather, electron-hole wavefunction overlap calculations in Ref. 16 only reveal, as the authors note, how two emissions (red and green) might be able to co-exist in the same nanostructure.

We find it more insightful and significant to distinguish between Auger recombination processes that take place in the core and those that are active in the arms. This distinction has not yet been made in the literature, although it is now clear that the two processes together determine PL properties in these complex seeded nanostructures. Specifically, via an experimental study, we are able for the first time to prove that red-PL blinking properties are

determined by the extent to which Auger is suppressed in the tetrapod core, while the key structural parameter for suppression of Auger in the arms is in fact arm length, rather than simply “volume” as previously suggested. In support of the former and to provide context, we now make an analogy to thick-shell QDs and add a **new reference (48)** that discussed theoretically the possible mechanisms responsible for the observed extreme suppression of Auger in the case of these QDs. We note, however, that a conclusive explanation for Auger suppression remains elusive, even for the case of the well-studied QDs systems. Therefore, we feel that such a treatment is beyond the scope of the current work. In support of the latter (enhanced Auger suppression in tetrapod arms), we provide **new reference 49**, which discusses how Auger recombination can be minimized in the transition from 0D (QD-like) to 1D (rod or wire-like) structures, i.e., in our analogy we compare the tetrapod arms to literature 1D structures.

Also, with respect to two-color multi-excitonic red and green emission, we now provide a quantitative assessment of the processes active in the arms – and their interplay with the core – that lead to this unique type of dual emission. We believe that this new addition to the manuscript speaks to the reviewer’s need for better “support” for our observations and indeed substantially elevates the work.

Specifically, we now provide a quantitative assessment of the actual number of excitons in the core as a function of pump fluence, as well as an assignment of their origin, i.e., whether they originated in the core or in the arms and were transferred to the core. This type of “exciton bookkeeping” allows us to identify the mechanism for dual emission. First, the number of excitons in the core was determined by analysis of the TA data according to a procedure described in **new reference 45**. Specifically, the average number of excitons per CdSe core at different excitation fluences was determined from the ratio of the signal magnitude at the peak and at long delays where only the radiative decay is contributing to the relaxation (**a/b ratio in new Figure 5b per new ref. 45**). Knowing the absorption cross-section of the core (calculated using **new equation shown on page 10 of revised SI**), the number of e-h pairs generated directly in the core for a given fluence could be calculated (tabulated in **new Table S4**). By difference, the number of e-h pairs in the core – but originating in the arms – could be determined (**new discussion/results on pages 17-19**).

Similarly, knowing the absorption cross-section of the arms (calculated using **new equation shown on page 10 of revised SI**), the number of e-h pairs generated in the arms for a given fluence could be calculated (tabulated in **new Table S4**). By comparing the number of e-h pairs generated in the arms with the number transferred to the core, it was possible to assess the efficiency of the arm-to-core “antenna” effect. This analysis is now provided on **page 17** of the revised manuscript, with the concluding statement:

“Using this approach, we find that at the high fluence of $116 \mu\text{J}/\text{cm}^2$ per pulse, for example, the number of electron-hole pairs in the core is ~ 2.5 on average. This is a small fraction of electron-hole pairs generated in the tetrapod as a whole at this pump fluence (~ 200 electron-hole pairs, Table S4). Thus, this result indicates that the vast majority ($\sim 99\%$) of excitons generated in the arms do not reach the core.”

That said, our new results clearly show the important contribution of arm excitations to core processes (**page 17**):

“Significantly, even though only $\sim 1\%$ of the electron-hole pairs generated in the arms reach the core, they push the core into the multiexciton regime and dictate the core relaxation dynamics.”

Also from analysis of TA data, we determine the process by which e-h pairs are transferred to the core, namely (**page 19, revised manuscript**):

“...the CdSe core is not populated by a transfer or migration of the band-edge exciton from the CdS arm, but, rather, that the fraction of electron-hole pairs generated within the arms relaxes directly into the CdSe core during intraband cooling (i.e., from high-energy arm excited states to band-edge core states).”

We further determine the fate of the e-h pairs in the arms that do not relax to the core states (**pages 19-20**):

“What is the fate of the rest of the carriers formed in the CdS arms? This question can be addressed by considering, for example, the results for low power excitation, i.e., $13 \mu\text{J}/\text{cm}^2$ per pulse. At this pump fluence, ~ 20 excitons are generated on average in a tetrapod (see Supplementary Information) but no green PL is observed. According to TA data < 1.5 excitons on average relaxes through the core (Figure 5b, main panel, green trace). This means that the majority of the generated excitons relax in the arms, as confirmed by the large CdS bleach observed in TA spectra at ~ 470 nm (Figure 5a, main panel). However, no arm emission is observed in the single-tetrapod spectra at this pump fluence, which indicates that the arm excitons decay only through non-radiative channels.

To determine relative contributions from possible non-radiative channels – carrier trapping and Auger recombination – we consider both the appearance of the bleach signal and the bleach relaxation kinetics.

First, the typical signatures of carrier traps (broad tail/band to the red of the main bleach) do not significantly contribute to the TA spectra (Figure 5a, main). As shown previously, in the case of cadmium chalcogenides the TA signal is mostly sensitive to electron dynamics, rather than hole dynamics.^{46,47} Therefore, in further analysis we will assume that this observation pertains primarily to electron traps. Second, the TA bleach relaxation dynamics of the excitons in the arms (Figure 5b, inset) for the 13 $\mu\text{J cm}^{-2}$ per pulse data (green trace) show ~20% drop in signal intensity within 1 ns. This fast relaxation is attributed to the decay of carriers through non-radiative channels, i.e., trapping and/or Auger recombination. Since even at this low fluence ~20 excitons are generated on average in the arms of a tetrapod, the Auger process likely contributes, at least in part, to the observed fast relaxation. That means that <20% of the relaxation can be attributed to an electron trapping process. However, as no arm PL (proportional to the product of the number of electrons and holes) is observed at this fluence, this result implies that non-radiative relaxation is dominated by hole trapping, whereby band-edge electrons recombine non-radiatively with the trapped holes.”

Thus, we are able to conclude that two-color multi-excitonic emission follows processes of hole-trap filling in the CdS arms and partial state-filling in the core. **Page 20:**

“The above analysis indicates that the observation of dual red and green emission in CdS/CdSe tetrapods requires at least partial filling of hole traps in the arms and partial state-filling in the CdSe core. With the two relaxation channels partially or completely blocked (e.g., through state filling or electrostatic repulsion) radiative decay in the arms becomes kinetically competitive and both red core emission and green arm emission are observed. The requirement of multi-level (but not complete) filling of the CdSe core states implies that the observation of dual emission is likely only when both the core and the arms are populated with multiple excitons. Since in this regime radiative decay must compete with the non-radiative Auger process, the optimization of the dual PL requires effective suppression of Auger recombination in both the core and the arms of the tetrapod.”

*In this way, we believe our experimental study affords important and novel fundamental mechanistic insight that qualifies for publication in *Nature Communications*.

Comment: In general, an aspect on which the authors should have worked more extensively is the evaluation of the number of excitons per nanoparticle in their experiments. Absorption cross section are known for these nanomaterials, which are actually out in the community since almost a decade (ref.1).

Response: In the revised manuscript, we provide an evaluation of the number of excitons per nanoparticle in new **Table S4 and Figure S1**, where Figure S1 also shows intensity vs. pump fluence and excited e-h pairs (i.e., PL saturation curves) from single tetrapods for each of the tetrapod structures. The number of excitons formed in each tetrapod was estimated based on excitation fluence and tetrapod absorption cross-section. We used the relation $N = j\sigma$, where N is the number of excitons generated, j is the per-pulse laser intensity in photons/cm² per pulse, and σ is the absorption cross-section of the tetrapod in cm².

The tetrapod and core-only absorption cross-sections were calculated, with volume as the basis, using known methods previously used for CdSe/CdS QDs (**new SI reference 12**). The absorption cross-section, σ_w , is calculated using the equation ,

$$\sigma_w = V\alpha_w |f_w|^2 \frac{n_w}{n_{\text{medium}}}$$

where V is the volume of the tetrapod, α_w is the absorption coefficient of CdS, f_w is a correction factor for local field effects, n_w is the refractive index of CdS, and n_{medium} is the refractive index of air. Further details of our approach can be found in the revised SI on page 10.

Comment: There is a good statistics (number of particles studied) for blinking experiments, but the statistics on the TEM measurements is instead not mentioned, on how many particles the measurements of size have been performed and what are the errors in the calculation of the volumes?

Response: We do in fact provide in the caption to Figure 1 the number of particles used to determine the tetrapod arm diameters, lengths and respective standard deviations: “...arm diameter (measured at base for 30-40 tetrapods in each case) and arm length...” For enhanced clarity, we have modified the caption further to say, “arm diameter (measured at the base of 3 arms for 30-40 tetrapods in each case) and arm length (also measured for 3 arms of 30-40 tetrapods for each of the tetrapod geometries)...” We hope this sufficiently addresses the reviewer’s concern.

With respect to tetrapod volumes, as stated in the manuscript (now page 8 in the revised manuscript) the volume of a truncated cone was used to “estimate” volume for TP1, TP3 and TP4, while the volume of a cylinder was used to estimate volume for TP2. Inputs for these volume calculations (arm diameters and arm widths) came from our measurements of TEM images, with one exception. Specifically, to calculate the volume of a truncated cone (i.e., a cone that ends with a flat tip rather than a point, which we believe more accurately represents the shape of our TP1, TP3 and TP4 tetrapods), both the large-end “base” radius and the small-end “tip” radius are required, along with cone height (arm length). We determined the values for base radius and cone height directly from TEM measurements. These quantities are provided, along with standard deviations generated by measuring 90-120 arms for each of the TP geometries as described above, in the caption for Figure 1. In contrast to arm base radius, we did not feel confident in our measurements of arm tip radius using TEM images, as the ends of the tetrapods often deviated in shape from that anticipated by an even progression of the cone from base to end. In particular, the very ends of the arms were often suddenly narrower than the segments of arm nearest to the tips. For this reason, we chose, instead, to measure the mid-point of the arms (again, using 90-120 arms). Assuming a roughly even progression from base to tip for the arm shape, the calculated tip radius was determined by following the lines created by connecting base-point and mid-point diameter lines (each perpendicular to and centered on a line representing arm length at arm end and middle, respectively) to the full length of the arm. Given that the actual tips are smaller than that estimated in this way, the calculated volumes are likely an overestimate of the actual volumes, but as the arms taper at a relatively even grade for the majority of the arm length, we feel that our approach provides a reasonable estimate of arm volume. However, given the simplifications that we have necessarily made with respect to arm shape, we do not feel that the volumes that we report warrant inclusion of statistics, i.e., inclusion of standard deviations, as doing so would imply greater confidence in these values than is appropriate due to the clear variability in shape from arm to arm and the deviation from perfect truncated cone geometry.

To clarify our approach and the results we report, we have now included the above explanation in Supplementary Information as a new section called, “**Calculating tetrapod volumes**” (SI pages 4-5), along with a table (**Table S2**) listing each of the relevant diameter and length parameters, and we have added a reference in the main text to this discussion.

Comment: Another aspect which I find well below the standards for publication in Nature Comm is the analysis of the ensemble data, what is the rationale for the normalization of the PLE spectra? This seems to be entirely arbitrary... what are the concentration of the solutions or films (not clear from the SI if these PLE measurements are performed in solution or films, while for time resolved it is clear that has been done on films).

Response: PLE was obtained from solutions. However, tetrapod solutions, or, more accurately, suspensions, were somewhat turbid (with the exception of TP2), causing scattering that lent uncertainty to absorption measurements. For this reason, we could not prepare or report definitively “equivalent-concentration” solutions based on absorption. For this reason, we compare PLE spectra as these are less affected by scattering.

The rationale for reporting the PLE spectra as we did is that by normalizing at a wavelength (energy) below the bandgap of the CdS arms (bulk bandgap: ~515 nm or 2.42 eV), i.e., where only the CdSe cores can absorb light and contribute to PL, we account for effects of different TP concentrations. In other words, we account for solution concentration by normalizing to the CdSe core PLE intensity, as we use identical cores to synthesize each of the four tetrapod geometries. Thus, for a given CdSe core PLE, we then compare PLE intensity (for red emission) from 400-515 nm, which includes contributions from the CdS arms, i.e., both CdSe core and CdS arms can absorb excitation photons in this spectral region and potentially contribute to PL. As we describe in the manuscript, we find, however, that PLE intensity trends do not necessarily follow absorption cross-section values, i.e., tetrapod volume trends. Therefore, the relative effectiveness of particular arm geometries in harvesting excitation energy for red core emission depends on more than size, cross-section or volume.

Because we are normalizing to a very weak signal (CdSe core PLE is weak relative to CdS arm PLE), a quantitative assessment of the arm-to-core carrier/exciton transfer process using the obtained PLE data is likely to have large associated errors. For this reason, we now include TA experimental results and their analysis, which allows a more quantitative determination of the number of excitons in the core that originated in the arms. This was only possible for TP2 tetrapods, however, as these were the only substantially non-scattering sample for which we were sufficiently confident in the TA results. To provide the qualitative picture of an imperfect arm-to-core antenna effect, we have retained the PLE data, but we have moved the PLE figure to the SI (Figure S6) and have included the above new discussion for clarification.

Comment: There is some speculation on the role of tapered arms and cylindrical. This is, however, not substantiated with a quantitative discussion.

Response: We agree with the reviewer on this point, and given the additions we have made in the revised version of the manuscript that now allow us to better interpret the differences between the tetrapod structures with respect to arm to core exciton funneling, as well as our more clear focus on blinking studies, we have removed these comments from the manuscript.

Comment: What is the exciton Bohr radius for bulk CdS?

Response: As we have removed discussion of exciton funneling related to TP shape effects, this question is no longer relevant.

Comment: How the authors expect the electron and hole wave functions are spread in the CdS upon excitation at 405 nm?

Response: This question is related to the reviewer's previous query regarding theoretical modeling for exciton wavefunction overlap. We again note that Ref. 16 provides a figure that purports to address this question, while for our work, we believe that our experimental study now combining PL, transient PL and TA analyses affords a clear picture of how arm-generated excitons lead either to the formation of core-located excitons or to excitons associated with the arm.

Comment: The presentation could be improved. The manuscript is very long and I believe it is too long because the authors are commenting too much on results from previous papers out in the literature. The flow is often interrupted with comparisons with giant dots that could be group in a final section discussing the results in a comprehensive picture. While it is too long and with far too many references to only partially related work, it misses the explanation of those concepts that are crucial for a broad readership to understand. Auger recombination and multi-excitons states, which are clearly two of the most important aspects of this work are not introduced properly with an explanation of what they are and how important they are for the photophysics of semiconductor nanocrystals.

Response: We agree and have substantially edited the revised manuscript for brevity where necessary and with the above described new clarifying/enhancing data and analyses. Also, we now define exciton and multiexciton in the text (**page 3 and 4, respectively**) and Auger recombination in a footnote (**new Reference 17 on page 5**). We further provide additional explanation of the key concepts pertaining to dual emission(s) in CdSe/CdS tetrapods on **pages 3-5**. To improve and simplify the presentation, we have also removed some comments related to giant dots, e.g., **page 8** regarding Stokes shift.

REVIEWERS' COMMENTS:

Reviewer #1 (Remarks to the Author):

comments to Editors only given at this stage

Reviewer #2 (Remarks to the Author):

I have reviewed the revised manuscript and reply to the comments carefully. I have found that the authors have provided a lot of more details and fundamental analysis. I felt the paper is ready to be accepted now.

Reviewer #3 (Remarks to the Author):

The authors have worked extensively on the manuscript and now include additional material. The results on blinking are analysed in greater detail and the manuscript is generally better presented. I am in strong disagreement with some of the statements made by authors on the novelty of dual emission at the single nanostructure level. Dual green and red PL emission in single tetrapods has been reported in Phys. Rev. B 82, 081306 (2010) and also discussed in relation to arm length. I suggest the authors to give a fair account of what has been done on these materials.

REVIEWER'S COMMENT:

Reviewer #3 (Remarks to the Author):

Comment: The authors have worked extensively on the manuscript and now include additional material. The results on blinking are analysed in greater detail and the manuscript is generally better presented. I am in strong disagreement with some of the statements made by authors on the novelty of dual emission at the single nanostructure level. Dual green and red PL emission in single tetrapods has been reported in Phys. Rev. B 82, 081306 (2010) and also discussed in relation to arm length. I suggest the authors to give a fair account of what has been done on these materials.

Response: We respectfully disagree with the reviewer. The referred to work, Phys. Rev. B 82, 081306 (2010), comprises 2 figures that show experimental data (a third shows the results of calculations). As suggested by the reviewer, the first experimental figure does show results for tetrapods possessing two different arm lengths; however, these are ensemble measurements. The second experimental figure is a single-nanostructure result, but it is simply a steady-state PL spectrum taken for a single tetrapod *representing one arm length*, along with polarization angle dependence of the emission. A summation of the polarization data is provided in the text for 20 tetrapods. The data were obtained at cryogenic temperatures. Furthermore, in all of this work, only a single pump fluence was reported, and this was described for the ensemble measurements as yielding an excitation level of only ~ 0.1 excitons per tetrapod, i.e., low pump fluence. In contrast, our report describes single-nanostructure behavior for four different tetrapod geometries at room temperature, does so for multiple pump fluences (the values of which are clearly presented; to ascertain the pump-dependent evolution of dual-emission behavior at the single-nanostructure level), and affords an unprecedented analysis of time-dependent photoluminescence behavior in terms of blinking and photobleaching. Thus, our work is clearly novel compared to Phys. Rev. B 82, 081306 (2010) with respect to showing and understanding dual emission at the single nanostructure level.

That said, despite the numerous differences between our work and the earlier report, per the editorial requests and per the editorial policies of Nature Communications, we have removed terms like "for the first time" to describe aspects of our work. In this way, as well, the issue raised by the reviewer has been resolved. We still do not include this reference, however, in our work as we, instead, include Ref. 16. Mauser et al. Phys. Rev. B 82, 081306 (2010) was published with almost the same author list as Ref. 16 and in the same year (overlapping authors between the two publications: Mauser, Da Como, Rogach, Huang, Talapin and Feldmann). Unfortunately, though, the two publications appear to provide contradictory mechanisms for dual emission. While Ref. 16 describes dual green and red emission as an Auger-dependent multiexcitonic emission process that can happen "*once the core levels are occupied with a high density of excitons... (to) create an exciton blockade where the electrons and holes, primarily photogenerated in CdS... are not subjected to a driving force for relaxing into the core and eventually experience efficient radiative recombination,*" Mauser et al. suggest that hole trapping in the CdS arm to a defect state and subsequent Coulomb drag bringing the electron to the hole affords the two color emission. We did not feel it appropriate to point out the apparent contradictions between the two works in our report. Also, as our report describes a clear pump-fluence dependency for the dual-emission process, we site Ref. 16 as being more consistent with our observations, though as described in the text, with clear distinctions.